



# 1-km monthly temperature and precipitation dataset for China from 1901–2017

Shouzhang Peng[1], Yongxia Ding[2], Zhi Li[3]

[1] State Key Laboratory of Soil Erosion and Dryland Farming on the Loess Plateau, Northwest A&F University, Yangling,
712100, China
[2] School of Geography and Tourism, Shaanxi Normal University, Xi'an, 710169, China
[3] College of Natural Resources and Environment, Northwest A&F University, Yangling, 712100, China

*Correspondence to*: Zhi Li (lizhibox@nwafu.edu.cn  szp@nwafu.edu.cn)

**Abstract:** High-spatial-resolution and long-term climate data are highly desirable for understanding climate-related natural
processes. China covers a large area with a low density of weather stations in some regions, especially in mountainous regions.
This study describes a 0.5' (~1 km) dataset of monthly air temperatures at 2 m (minimum, maximum, and mean TMPs) and
precipitation (PRE) for China from 1901–2017. The dataset was spatially downscaled from 30' climatic research unit (CRU)
time series dataset with the climatology dataset of WorldClim by using Delta spatial downscaling and evaluated using
observations during 1951–2016 from 496 weather stations across China. Moreover, the bicubic, bilinear, and nearest-neighbor
interpolation methods were compared in the downscaling processes. Among the three interpolation methods, bilinear
interpolation exhibited the best performance to generate the downscaled dataset. Compared with the evaluations of the original
CRU dataset, the mean absolute error of the new dataset (i.e., 0.5' downscaled dataset with the bilinear interpolation) relatively
decreased by 35.4 %–48.7 % for TMPs and 25.7 % for PRE, the root-mean-square error relatively decreased by 32.4 %–44.9 %
for TMPs and 25.8 % for PRE, the Nash–Sutcliffe efficiency coefficients relatively increased by 9.6 %–13.8 % for TMPs and
31.6 % for PRE, and the correlation coefficients relatively increased by 0.2 %–0.4 % for TMPs and 5.0 % for PRE. Further,
the new dataset could provide detailed climatology data and annual trend of each climatic variable across China, and the results
could be well evaluated using observations at the station. Although the evaluation of new dataset was not carried out before
1950 owing to a lack of data availability, the downscaling procedure used data from CRU and WordClim and did not
incorporate observations. Thus the quality of the new dataset before 1950 mainly depended on that of the CRU and WordClim
25  datasets. The evaluations showed that the overall quality of the CRU and WordClim datasets was satisfactory, and the
downscaling procedure further improved the quality and spatial resolution of the CRU dataset. The new dataset will be useful
in investigations related to climate change across China. The dataset presented in this article has been published in Network
Common Data Form (NetCDF) at http://doi.org/10.5281/zenodo.3114194 for precipitation (Peng, 2019a) and
http://doi.org/10.5281/zenodo.3185722 for air temperatures at 2 m (Peng, 2019b). The dataset includes 156 NetCDF files
30  compressed with zip format and one user guidance text file.



## 1 Introduction

High-spatial-resolution and long-term climate data are required for accurate investigations of changes in climate and climate-related phenomena that affect hydrology, vegetation cover, and crop production (Gao et al., 2018; Caillouet et al., 2019; Peng et al., 2018; Peng and Li, 2018). Although meteorological observation networks are increasingly incorporating data from a greater number of weather stations and contributions from an increasing number of governments and researchers around the world, observation networks still suffer from a low station density and spatial resolution (Caillouet et al., 2019; Peng et al., 2014), especially in mountainous areas (Gao et al., 2018). The installation and maintenance of weather stations in mountainous areas are challenging (Rolland, 2003). Accordingly, several interpolation methods such as inverse distance weighting, kriging methods, and regression analysis are usually used to generate meteorological data for those ungauged areas (Li et al., 2010; Li et al., 2012; Zhao et al., 2004; Atta-ur-Rahman and Dawood, 2017; Peng et al., 2014). However, the accuracy of the results of these interpolation methods depends on station density (Gao et al., 2018; Peng et al., 2014). Therefore, it is necessary to use climatic proxy data to generate long-term and high-spatial-resolution climate data.

Proxy monthly temperature (TMP) and precipitation (PRE) data products are released by several climate research organizations, such as the General Circulation Models (GCMs) of Intergovernmental Panel on Climate Change (Brekke et al., 2013), the Climatic Research Unit (CRU) of the University of East Anglia (Harris et al., 2014), the Global Precipitation Climatology Centre (GPCC) (Becker et al., 2013), and Willmott & Matsuura (W&M) (Matsuura and Willmott, 2015a, b). These products have a long time series (> 100 years) and moderate spatial resolution ($\geq$ 30'). Compared with GCM products, CRU, GPCC, and W&M products are generated from the data obtained from observational stations, and thus, they have higher reliability. Furthermore, compared with GPCC and W&M products, CRU products include several TMP and PRE variables such as monthly mean TMP, maximum TMP, minimum TMP, and PRE. Therefore, CRU products have been widely employed to investigate climate effects globally (Kannenberg et al., 2019; Lewkowicz and Way, 2019; Bellprat et al., 2019). Although CRU products offer the advantage of reflecting long-term climate effects, their spatial resolution (30', approximately 55 km) limits their ability to reflect the effects of complex topographies, land surface characteristics, and other processes on climate systems (Xu et al., 2017; Peng et al., 2018). This also prevents CRU data from providing realistic and reliable climate change information on fine scales, which is imperative when developing adaptation and mitigation strategies that are suitable for use on local scales (Giorgi et al., 2009; Peng et al., 2019). Therefore, it is necessary to spatially downscale and correct CRU climate data.

Previous studies have shown that the Delta downscaling framework performs well in downscaling climate data (Mosier et al., 2014; Peng et al., 2018; Peng et al., 2017; Wang and Chen, 2014; Brekke et al., 2013). This framework uses a low-spatial-resolution monthly time series data and high-spatial-resolution reference climatology data as inputs. The high-spatial-resolution climatology data must be physically representative and have fine-scale distribution of meteorological variables over the landscape of interest (Mosier et al., 2014; Peng et al., 2017). As a result of incorporating high-spatial-resolution reference climatology data, downscaled results often have higher accuracy compared with that of original data with respect to weather



station data, especially monthly mean TMP and PRE (Peng et al., 2018). Thus, the Delta downscaling framework can downscale and correct low-resolution climate data.

China has a large area and includes many mountainous areas. Even the establishment of additional weather stations has not made it possible to fully satisfy requirements for long-term, high-spatial-resolution climate data, especially at finer geographical scales and for mountainous areas. Furthermore, most weather stations in China were established after 1950, and thus, long-term observational climate data are insufficient (Peng et al., 2018). These above shortcomings limit the types of studies that can be conducted on long-term climate change and effects of climate change at fine geographical scales across China.

The objective of this study was to generate a long-term climate dataset having a high spatial resolution for China by downscaling CRU time series data with high-spatial-resolution reference climatology dataset. The specific climate data types generated included monthly air TMPs at 2 m (mean, maximum, and minimum TMPs) and PRE variables with a spatial resolution of 0.5' (approximately 1 km) from January 1901 to December 2017. First, the reference climatology data with different spatial resolutions and the 30' original CRU time series data were evaluated through observations. Second, the 30' original CRU time series data were spatially downscaled to four spatial resolutions (e.g., 10', 5', 2.5', and 0.5'), corresponding to the spatial resolutions of the reference climatology data by using the Delta downscaling framework. The downscaled data were validated through observations. In addition, the accuracy of the 0.5' downscaled data was compared with that of the downscaled data with other spatial resolutions, in order to demonstrate the performance of the downscaling framework and 0.5' downscaled data. Finally, the climatology data and annual trends in TMPs and PRE were investigated using the 30' original CRU, 0.5' downscaled, and observed data to demonstrate the performance of the 0.5' downscaled data.

## 2 Data

### 2.1 CRU time series data

The monthly mean, maximum, minimum air TMPs at 2 m, and PRE were obtained for January 1901 to December 2017 with a spatial resolution of 30'. These variables were obtained from the CRU TS v. 4.02 dataset (http://www.cru.uea.ac.uk) (Harris et al., 2014). Methodologies used by the CRU group to construct 30' time series dataset are similar to the Delta downscaling framework employed herein (see section 3.1). First, more than 5000 weather stations were employed, and each station series was converted to anomalies by subtracting (for temperatures) or dividing (for precipitation) the 1961−1990 normal from the station's data. Then, the station anomaly time series data were linearly interpolated into 30' grids covering the global land surface. Finally, the grid anomaly time series data were transformed back to absolute monthly values by using 30' reference climatology dataset during 1961−1990. Specifically, the 30' reference climatology dataset used by the CRU group contained the climatology data for each month and was obtained from New et al. (1999). This climatology data were generated by a function considering the latitude, longitude, and elevation, based on 3615–19800 weather stations located globally. Elevation data used in this climatology dataset had a spatial resolution of 30', which was a mean result of the global 5' digital elevation





model. Specifically, elevation at each 30' grid was the mean of 36 grids of the 5' digital elevation model (New et al., 1999). Therefore, the CRU dataset could represent orographic effects on climate variation at 30' spatial resolution. Compared with similar gridded products, the CRU dataset exhibited better performance. In addition, 323 weather stations across China region were employed by CRU group to generate CRU time series data (Harris et al., 2014) (Fig. 1).

**2.2 WorldClim data**

To downscale CRU TMPs and PRE time series data to higher spatial resolutions, we obtained four high-resolution reference datasets at spatial resolutions of 10', 5', 2.5', and 0.5' from WorldClim v. 2.0 (http://worldclim.org) (Fick and Hijmans, 2017). The reference datasets consisted of monthly averages of climatic variables (mean, maximum, and minimum air TMPs at 2 m, as well as PRE) for 1970–2000, generated based on 9000–60000 weather stations located globally, by using the thin-plate
splines interpolation method. Thus, for each climatic variable, it had 12 climatology layers, representing climatology data ranging from January to December. Remarkably, the interpolation considered covariation with latitude, longitude, elevation, distance to the nearest coast, and three satellite-derived covariates: the maximum and minimum land surface temperature and cloud cover, obtained from the MODIS satellite platform. Thus, these reference data reflected orographic effects and observed climate information for each month. Moreover, cross-validation correlations indicated that these reference data exhibited good
performance globally because of the introduction of satellite-derived and distance to the nearest coast covariates. In addition, weather stations over China used in the WorldClim were the same as those used in the CRU group (Fick and Hijmans, 2017) (Fig. 1). For an independent evaluation of the downscaled dataset in this study, these weather stations were excluded.

**2.3 Observations**

To evaluate the performance of the downscaling procedure, the observed long-term monthly TMPs (i.e., mean, maximum,
minimum air TMPs at 2 m) and PRE variables across China were obtained from the National Meteorological Information Center of China (http://data.cma.cn/en). This dataset included observations from 496 national weather stations (Fig. 1) during 1951–2016. These stations were not taken part in the generations of CRU time series and WorldClim data. Figure 2 shows the orographic statistic information (e.g., elevation, slope, and aspect) of China and the 496 independent weather stations. The results indicated that the proportion of independent weather station in different orographic gradients almost corresponded to
that in China, except in areas with elevations exceeding 4500 m, which indicated that these weather stations could represent climate variation over China and be used for validating the downscaled dataset. This exception is inevitable because of the observability, installation, and maintenance of weather stations over those areas. In addition, although China had a few weather stations during 1901–1950, all these stations have been used to generate CRU time series data before 1950. Therefore, this study would evaluate the downscaled dataset during 1951–2016 by using 496 independent and representative stations.





## 3 Methods

### 3.1 Spatial downscaling

Delta downscaling was employed to generate monthly TMPs and PRE for the period 1901–2017 at spatial resolutions of 10', 5', 2.5', and 0.5'. The Delta downscaling framework used in this study includes the following four steps (Peng et al., 2018):

First, a climatology dataset was constructed for each month and each climatic variable based on 30' CRU time series. In this step, the annual average at each month for TMPs (i.e., mean, maximum, and minimum TMPs) and PRE variables were constructed based on CRU TMPs and PRE time series data. Specifically, the constructed climatology dataset had a spatial resolution of 30', which is the same as the CRU dataset. Moreover, to match the period of high-resolution reference datasets from WorldClim, the 30' climatology dataset was constructed for the period 1970–2000. Thus, for each climatic variable, the

dataset would have 12 climatology layers during 1970–2000 with a spatial resolution of 30'.

Second, the 30' anomaly time series data were derived for each climatic variable based on the 30' CRU time series data and the constructed climatology dataset. In this step, the TMP anomaly time series data were calculated as the difference between the TMP time series and the TMP climatology data in the corresponding month, while the PRE anomaly time series data were calculated as the ratio of the PRE time series to the PRE climatology data in the corresponding month. The specific calculation

equations are introduced as follows:

$$\text{An\_TMP}(yr, m) = \text{TMP}(yr, m) - \text{CRUClim\_TMP}(m) \qquad (1)$$

$$\text{An\_PRE}(yr, m) = \text{PRE}(yr, m) / \text{CRUClim\_PRE}(m) \qquad (2)$$

where An_TMP($yr$, $m$) and An_PRE($yr$, $m$) are the anomaly for temperatures and precipitation, respectively, at $m$ month and $yr$ year; TMP($yr$, $m$) and PRE($yr$, $m$) are the absolute temperatures and precipitation values, respectively, at $m$ month and $yr$

year; CRUClim_TMP($m$) and CRUClim_PRE($m$) are the 30' climatology for temperatures and precipitation, respectively, at $m$ month. $m$ ranges from January to December.

Third, the 30' anomaly time series dataset was spatially interpolated to a higher spatial resolution. In this step, the 30' anomaly grids at each time step are interpolated to four spatial resolutions (i.e., 10', 5', 2.5', and 0.5') to match the spatial resolutions of the reference datasets from WorldClim. Specifically, three interpolation methods are employed in this step,

including bicubic interpolation, bilinear interpolation, and nearest-neighbor interpolation methods. This study would compare the performances of these methods to select a reasonable interpolation approach.

Finally, the high-spatial-resolution anomaly time series dataset was transformed to an absolute climatic time series dataset based on the reference datasets from WorldClim at the corresponding spatial resolutions. In this step, the anomaly is undone at each time. Therefore, addition is used for TMPs, while multiplication is used for PRE. The specific calculation equations

are introduced as follows:

$$\text{TMP}(yr, m, res) = \text{An\_TMP}(yr, m, res) + \text{WorldClim\_TMP}(m, res) \qquad (3)$$

$$\text{PRE}(yr, m, res) = \text{An\_PRE}(yr, m, res) \times \text{WorldClim\_PRE}(m, res) \qquad (4)$$





where *res* represents the spatial resolution, i.e., 10', 5', 2.5', and 0.5'; TMP(*yr, m, res*) and PRE(*yr, m, res*) are the absolute temperatures and precipitation values with a spatial resolution of *res*, respectively, at *m* month and *yr* year; An_TMP(*yr, m, res*) and An_PRE(*yr, m, res*) represent anomalies with a spatial resolution of *res* for temperatures and precipitation, respectively, at *m* month and *yr* year; WorldClim_TMP(*m, res*) and WorldClim_PRE(*m, res*) represent climatology dataset

from WorldClim at a spatial resolution of *res* for temperatures and precipitation, respectively, at *m* month.

To visually present the downscaling processes, Figure 3 illustrates the components and steps of the Delta downscaling framework for obtaining the mean TMP by using the CRU 30' time series and WorldClim 0.5' climatology dataset. Specifically, to effectively interpolate the 30' anomaly time series dataset in China and conveniently implement the downscaling processes in the program code, downscaling was carried out in a rectangular region covering China (Fig. 3).

**3.2 Evaluation criteria**

Four statistic indexes were used to evaluate the original CRU and downscaled datasets. The indexes included Pearson's correlation coefficient (Cor), mean absolute error (MAE), root-mean-square error (RMSE), and Nash–Sutcliffe efficiency coefficient (NSE). The Cor was used to evaluate the correlation between original/downscaled and observed values. The MAE and RMSE assessed the bias between original/downscaled and observed values by using Eqs. (5) and (6). The NSE was used

to evaluate the performance of original and downscaled datasets by using Eq. (7). The NSE ranges from 1 (best fit) to negative infinity (worst fit) (Nash and Sutcliffe, 1970).

$$\text{MAE} = \frac{1}{n} \sum_{i=1}^{n} |P_i - O_i| \qquad (5)$$

$$\text{RMSE} = \sqrt{\frac{1}{n} \sum_{i=1}^{n} (P_i - O_i)^2} \qquad (6)$$

$$\text{NSE} = 1 - \frac{\sum_{i=1}^{n} (P_i - O_i)^2}{\sum_{i=1}^{n} (O_i - \bar{O})^2} \qquad (7)$$

where $P_i$ is the original or downscaled value, $O_i$ is the observed value, and *n* is the number of records.

**4 Results**

**4.1 Evaluation of WorldClim data at different spatial resolutions**

This study evaluated the reliability of the WorldClim dataset employed in this study based on observations from independent weather stations. Overall, the monthly climatology data with respect to temperatures and precipitation exhibited a high

performance to represent the monthly climatology data over China region during 1970–2000, and the climatology dataset exhibited good performance at a higher spatial resolution. Specifically, the absolute errors of the WorldClim datasets decreased with increasing spatial resolution (Table 1) and the correlations to the observations increased with increasing spatial resolution (Table 2), especially for the 0.5' WorldClim dataset. Thus, the WorldClim datasets employed in this study could be used as an input for the downscaling processes carried out in this study.





## 4.2 Evaluation of original CRU temperatures and precipitation data

Before downscaling, this study evaluated the performance of the original CRU time series dataset employed in this study. Table 3 presents the averaged evaluation over independent weather stations for the original monthly TMPs and PRE variables in the time series (1951–2016) based on independent observations. The results show that (1) the dataset exhibited good performance with respect to determining the original monthly TMPs and PRE values; (2) the performance of the NSE and Cor indexes in evaluating the values of TMPs was better than that for evaluating the value of PRE. Specifically, the MAE of the minimum, mean, and maximum TMPs, as well as of PRE were 1.766 °C, 1.598 °C, 2.034 °C, and 17.85 mm, respectively; the RMSE of the minimum, mean, and maximum TMPs, as well as of PRE, were 1.947 °C, 1.759 °C, 2.206 °C, and 29.559 mm, respectively; the NSE of the minimum, mean, and maximum TMPs, as well as of PRE, were 0.887, 0.888, 0.8, and 0.614 respectively; and the Cor of the minimum, mean, and maximum TMPs, as well as of PRE, were 0.994, 0.996, 0.995, and 0.885, respectively.

Figure 4 maps the MAE of the original TMPs and PRE variables at each independent weather station. The results show that (1) the original TMPs had larger biases in the northwest of China, especially at high-elevation regions and the Qinghai–Tibet Plateau; and (2) the original PRE had greater biases in the southern part of Qinghai–Tibet Plateau and China.

## 4.3 Validation of downscaled CRU temperatures and precipitation data

Table 3 presents the averaged evaluation over independent weather stations for the downscaled monthly TMPs and PRE in the time series (1951–2016) at different spatial resolutions. The results show that (1) compared with the original dataset, the downscaled dataset had lower MAE and RMSE values and higher NSE values; (2) the increased spatial resolution of the WorldClim reference dataset from 10' to 0.5' resulted in decreased MAE and RMSE values and increased NSE values; (3) of the three interpolation methods employed in the Delta downscaling framework, the downscaled data using the bilinear interpolation method had the lowest MAE and RMSE values and highest NSE values at each spatial resolution; and (4) the performance of the Delta downscaling framework was better for TMPs than PRE. Specifically, compared with the original dataset, the MAE of the downscaled minimum TMP at 0.5' under the bilinear interpolation method decreased to 1.05 °C (relative decrement of 35.4 %), the RMSE decreased to 1.248 °C (relative decrement of 35.9 %), the NSE increased to 0.972 (relative increment of 9.6 %), and the Cor increased to 0.998 (relative increment of 0.4 %). For the mean TMP, the MAE of the downscaled data at 0.5' under the bilinear interpolation method decreased to 0.820 °C (relative decrement of 48.7 %), the RMSE decreased to 0.969 °C (relative decrement of 44.9 %), the NSE increased to 0.981 (relative increment of 10.5 %), and the Cor increased to 0.998 (relative increment of 0.2 %). For the maximum TMP, the MAE of the downscaled data at 0.5' under the bilinear interpolation method decreased to 1.282 °C (relative decrement of 37.0 %), the RMSE decreased to 1.491 °C (relative decrement of 32.4 %), the NSE increased to 0.91 (relative increment of 13.8 %), and the Cor increased to 0.997 (relative increment of 0.2 %). For PRE, the MAE of the downscaled data at 0.5' under the bilinear interpolation method relatively decreased by 25.7 %, the RMSE relatively decreased by 25.8 %, the NSE relatively increased by 31.6 %, and the





Cor relatively increased by 5.0 %. Overall, the downscaled datasets had higher accuracy than the original CRU dataset, especially for the 0.5' downscaled dataset with the bilinear interpolation method, which was, therefore, the new dataset proposed by this study.

Figure 5 maps the relative decrement of MAE from the 30' original dataset to the 0.5' downscaled dataset with the bilinear interpolation method. Compared with the MAE of the original dataset, the MAE of the downscaled dataset were lower in all independent stations, especially in the northwest of China and Qinghai–Tibet Plateau.

### 4.4 Climatology of China based on 0.5' downscaled dataset

Table 4 lists the averaged climatology data obtained from independent weather stations during 1951–2016, based on the 30' original dataset, the 0.5' downscaled dataset with bilinear interpolation, and observations. The annual mean temperature and total precipitation were used to represent the climatology data in terms of mean TMP and PRE, while 1 % and 99 % quantiles (Q1 and Q99) of the monthly minimum and maximum TMPs, respectively, were selected to represent climatology in terms of minimum and maximum TMPs. This is because quantile temperatures are more reliable than absolute minimum and maximum TMPs if an outlier exists. The results indicate that the averaged climatology data for each climatic variable from the 0.5' downscaled data was closer to that from the observed data than that from the 30' original data. Specifically, the averaged climatology differences between the 0.5' downscaled and observed data were -0.02 °C for the Q1 of monthly minimum TMP, -0.18 °C for the Q99 of monthly maximum TMP, 0.01 °C for the annual mean TMP, and -0.5 mm for the annual total PRE.

To further illustrate the ability of the downscaled data to reflect climatology, Figure 6 shows the box plots of climatology anomaly during 1951–2016 for the 30' original and 0.5' downscaled datasets at independent weather stations, where the climatology anomaly is equal to the bias from the original/downscaled data to the observed values at each station. The results show that the climatology anomaly from the 0.5' downscaled dataset more intensively embraced the 0 value than that from the 30' original dataset, especially for its median and mean values. These results imply that the 0.5' downscaled dataset with bilinear interpolation could represent climatology in TMPs and PRE of China, compared with the 30' original dataset.

In addition, we investigated climatology by using the 0.5' downscaled TMPs and PRE data generated by the bilinear interpolation method for 1901 to 2017 (Fig. 7). The value of Q1 of the minimum TMP for China ranged from -50.15 °C to 17.21 °C, with an average of -17.1 °C, and the lowest value corresponded to a location in the western part of the Qinghai–Tibet Plateau (Fig. 7a). The value of Q99 of the maximum TMP ranged from -16.33 °C to 42.27 °C, with an average of 26.88 °C, and the highest value was observed at a location in the Turpan Basin (Fig. 7b). The annual mean TMP ranged from -34.41 °C to 26.39 °C, with an average of 6.18 °C, and the lowest and highest values correspond to locations in the western part of the Qinghai–Tibet Plateau and Hainan Island, respectively (Fig. 7c). The mean annual total PRE ranged from 3.2 mm to 4854.0 mm, with an average value of 564.4 mm, and the minimum and maximum values correspond to locations in the northwestern part of the Qinghai–Tibet Plateau near the Tarim Basin and Taiwan Island, respectively (Fig. 7d). The climatology data for the three TMPs varies with the topography and notably decreases with orographic uplift. The climatology



data for PRE decrease from the southeastern coastal region to the northwestern region. These results almost fit the orographic and coast effects on the climatology of China.

## 4.5 Trends of the annual temperatures and precipitation in China

Figure 8 maps the annual trends in TMPs and PRE over China during 1951–2016 based on the 0.5' downscaled dataset with bilinear interpolation, the 30' original dataset, and the observed dataset. The results show that (1) the annual values of TMPs and PRE in the 0.5' downscaled dataset were closer to the observations than the original values in the time series; (2) the annual trends from the 0.5' downscaled dataset were closer to the observed trends than to the trends from the 30' original data; and (3) the time correlations between the 0.5' downscaled and observed data were slightly better than those between the 30' original and observed data, although the later were not bad. Furthermore, the annual trends in the TMPs in the 0.5' downscaled dataset were underestimated, while those in the PRE in the 0.5' downscaled dataset were overestimated. Specifically, there were underestimated by 0.053, 0.048, and 0.06 °C 10 yr⁻¹ for the minimum, maximum, and mean TMPs and overestimated by 0.505 mm 10 yr⁻¹ for the PRE. Overall, the 0.5' downscaled and observed data had minor differences with respect to annual trends and high time correlations, and thus, the 0.5' downscaled dataset can be used to represent temporal variations and trends in TMPs and PRE across China.

In addition, we investigated the spatial patterns of annual trends in TMPs and PRE from 1901 to 2017 across China by using the 0.5' downscaled dataset with bilinear interpolation (Fig. 9). A 95% significance level was selected to represent the significance of the trend for each climatic variable. The annual minimum TMP exhibited a significant upward trend, from 0.018 °C 10 yr⁻¹ to 0.240 °C 10 yr⁻¹, with an average of 0.131 °C 10 yr⁻¹, over areas accounting for approximately 94.17 % of the total land area of China (Figs. 9a and e). The annual maximum TMP exhibited a significant upward trend, from 0.016 °C 10 yr⁻¹ to 0.171 °C 10 yr⁻¹, with an average of 0.081 °C 10 yr⁻¹, over areas accounting for approximately 80.85 % of the total land area of China (Figs. 9b and f). Meanwhile, the annual maximum TMP exhibited a significant downward trend, from 0.019 °C 10 yr⁻¹ to 0.034 °C 10 yr⁻¹, with an average of 0.027 °C 10 yr⁻¹, in areas accounting for only approximately 0.33 % of the land area of China (Figs. 9b and f). The annual mean TMP exhibited a significant upward trend, from 0.017 °C 10 yr⁻¹ to 0.189 °C 10 yr⁻¹, with an average of 0.104 °C 10 yr⁻¹, over areas accounting for approximately 90.92 % of the total land area of China (Figs. 9c and g). The annual PRE exhibited a significant upward trend, from 0.11 mm 10 yr⁻¹ to 21.206 mm 10 yr⁻¹, with an average of 3.306 mm 10 yr⁻¹, over areas accounting for approximately 22.02 % of the total land area of China (Figs. 9d and h). Meanwhile, the annual PRE exhibited a significant downward trend, from 0.13 mm 10 yr⁻¹ to 30.321 mm 10 yr⁻¹, with an average of 7.147 mm 10 yr⁻¹, over areas accounting for only approximately 2.01 % of China (Figs. 9d and h). Therefore, the 0.5' downscaled data with the bilinear interpolation proposed by this study can draw the detailed spatial variability of the trends in TMPs and PRE across China.



## 5 Data availability

The 0.5' downscaled dataset with bilinear interpolation developed in this study has been published in network Common Data Form (NetCDF) at http://doi.org/10.5281/zenodo.3114194 for precipitation (Peng, 2019a) and http://doi.org/10.5281/zenodo.3185722 for air temperatures at 2 m (Peng, 2019b). The dataset includes the monthly minimum, maximum, and mean temperatures, as well as the monthly total precipitation from January 1901 to December 2017. Because of the availability of original CRU data and the spatial resolution of the reference climatology data, the data covers most of the land area of China, with a geographic range of 18.2–53.5° N and 73.5–135.0° E. The total number of grids is 13,808,747. To reduce the size of the NetCDF file, the data for each climatic variable are divided into intervals of 3 years. TMPs and PRE are expressed to precisions of 0.1 °C and 0.1 mm, respectively, and they are stored using int16 format. Thus, each file contains 36 months of data and requires 2.42 GB of storage space. This file size should be convenient for processing by modern computers, and subparagraph storage in the time series can satisfy needs for quick data access for a specific period. Each file name indicates the data contained in the file, in the format "data type"_"beginning year"_"ending year".nc. For example, the file named tmn_1901_1903.nc contains minimum temperature data from 1901 to 1903. The total number of NetCDF files is 156, and the disk usage of the dataset in nc format is approximately 378 GB. After compression in zip format, the size of each file is approximately 300 MB, and all the files occupy a total of 47.8 GB. This dataset will be updated yearly because the CRU TS dataset is updated yearly, and new data will be available for download from the website identified above.

The monthly TMPs and PRE data in the 30' original dataset from 1901 to 2017 were obtained from the CRU TS v. 4.02 dataset (http://www.cru.uea.ac.uk/data, last access: 25 Apr 2019). The high-resolution reference data at spatial resolutions of 10', 5', 2.5', and 0.5' for TMPs and PRE were supported by WorldClim v. 2.0 (http://worldclim.org/version2, last access: 25 Apr 2019). The observed monthly meteorological data from the 496 weather stations across China were provided by the National Meteorological Information Center of China (http://data.cma.cn/en, last access: 25 Apr 2019).

## 6 Discussion, limitations, and recommendations

Although the original CRU dataset with a 30' spatial resolution was not evaluated as being poor, the 0.5' downscaled data with bilinear interpolation was evaluated as being better, with deviations decreasing by approximately 35.4 %–48.7 % for TMPs and 25.7 % for PRE, relative to the original CRU data (Table 3). Thus, the original CRU dataset needs to be corrected. Many factors contribute to the deviations, such as observational errors, sample size, and operator errors in gathering the original CRU data. However, little work done to address this issue. Previous studies have indicated that topographic information (e.g., elevation, location, slope, and aspect) may be key factors in correcting deviations, especially in mountainous areas (Gao et al., 2018; Peng et al., 2014; Gao et al., 2017). Therefore, a high-resolution reference climatology dataset containing detailed topographic information, as well as the effects of distance to the nearest coast and satellite-derived covariates, was used in this study to downscale the 30' original CRU dataset to a 0.5' dataset consisting of monthly TMPs and PRE from January 1901 to December 2017 across China, which has a low density of weather stations in mountainous areas. To the best of our knowledge,





this 0.5' downscaled dataset is the first dataset (version 1.0) developed with such a high spatiotemporal resolution over such a long time period for China.

Compared with the original CRU dataset, the downscaled dataset exhibited smaller deviations and higher spatial resolutions, suggesting that the Delta downscaling framework can be used to downscale and correct low-spatial-resolution climate data.

This should be attributed to the introduction of the high-spatial-resolution WorldClim data, because the reference climatology dataset with higher spatial resolution could produce more accurate downscaled data with a higher spatial resolution (Tables 1–3). Remarkably, because of the introduction of the averaged 30' elevation information in the original CRU data, this data weakens the representation of TMPs and PRE in the actual land surface, especially in regions with complex terrain. Moreover, the original CRU dataset was evaluated at weather stations, which are often located in valleys near the county or city. Thus,

the TMPs and PRE from the CRU dataset exhibited lower and higher values than those from the observations, respectively (Table 4 and Figure 6). However, the deviations were decreased to a certain extent in the 0.5' downscaled dataset (Table 4 and Figure 6). Even so, the Delta downscaling processes did not improve the time correlations between 0.5' downscaled and observed data by a considerable extent (Table 3). This could be attributed to the fact that the Delta downscaling processes focus on correcting deviations and downscaling the spatial resolution, using the 12 climatology layers from the WorldClim

dataset. In the geographical space, the corrections are evident, especially in the northwest of China and Qinghai–Tibet Plateau (Figure 5), which should result from the introduction of orographic effects, distance to the nearest coast, and effects of satellite-derived covariates in the WorldClim dataset.

The 0.5' downscaled TMP and PRE dataset with bilinear interpolation captures the detailed climatology of the whole of China very well (Fig. 7). It accurately represents climate characteristics such as the minimum TMP at high elevations (e.g.,

the Qinghai–Tibet Plateau), the maximum TMP at low elevations (e.g., Turpan Basin), and heavy PRE in marine areas (e.g., Taiwan Island). The biases of the climatology data were only -0.02 °C for the minimum TMP, -0.18 °C for the maximum TMP, 0.01 °C for the mean TMP, and -0.5 mm for the PRE (Table 4). Furthermore, the climatology anomaly at each weather station from the 0.5' downscaled dataset is closer to 0 than that from the 30' original dataset (Fig. 6). The 0.5' downscaled dataset with bilinear interpolation also represents detailed annual trends in climatic variables over China very well (Fig. 9). The dataset

precisely represents the trends and their significance levels over the geographic space, such as significant increasing and decreasing trends for the maximum TMP and PRE. In general, compared with the 30' original dataset, this dataset captures the annual trends very well (Fig. 8); the 0.5' downscaled and observed data exhibited high time correlations and minor differences in annual trends (Fig. 8). Therefore, the 0.5' downscaled dataset with bilinear interpolation can be used successfully to assess climate change and its spatial effects across China.

As mentioned previously, the accuracy of the reference climatology dataset largely determines the quality of the dataset. In this study, the reference climatology dataset from WorldClim was adopted. Although the evaluation indicated that the quality of the dataset is very good, there is a gap between the dataset and observed data. We think that a new and better reference climatology dataset should be generated using the observed data gathered from across China. However, the current release of public climate data over China is insufficient to construct a better reference climatology dataset than that available from





WorldClim. In ongoing research, we are devoting efforts to collecting more public and private climate data so that we can construct a better reference climatology dataset and then generate a more accurate downscaled dataset for China.

Another limitation is the difficulty of validating new dataset before 1950. Although China had a few weather stations with data collected from 1901, all of them have been used to generate the CRU time series (Harris et al., 2014). Therefore, we

cannot verify data quality before 1950 because of a lack of data availability. However, the downscaling procedure used data from CRU and WorldClim datasets and did not incorporate the observation, the quality of new dataset before 1950 mainly depends on that of the CRU and WorldClim datasets. Furthermore, the evaluations showed that the overall quality of the CRU and WorldClim dataset is satisfactory, and the downscaling procedure can further improve the quality of the CRU dataset, as well as enhance its spatial resolution.

The usage of some evaluation indexes may have defects and should be clarified in this study. The involved indexes used in this study can be classified into two groups: one group based on the sums of squared errors (i.e., RMSE and NSE) and the other group based on the sums of error magnitudes (i.e., MAE). The sums of the squared errors are influenced by three independent variables, such as the mean of individual error magnitudes, variability among error magnitudes, and number of observations or domain of integration (Willmott et al., 2009). Willmott and Matsuura (2005) recommended MAE as an

evaluation criterion for estimations. However, this study adopted CRU time series dataset as a unique original dataset and the observations from 496 weather stations as a unique evaluation dataset. Thus, the variations in RMSE or NSE at different cases were only influenced by the mean of individual error magnitudes, which were introduced by different spatial resolutions and interpolation methods. Thus, the RMSE and NSE indexes satisfied the evaluation criteria of this study. Further, the evaluation indexes were mainly used to compare the performance of the downscaled and original datasets. Therefore, the usage of

evaluation indexes in this study is reasonable.

In addition, because of the limitations associated with the computational resources and the resolutions of reference climatology and original CRU dataset, the resolution of the new dataset is limited to monthly and a 0.5' (approximately 1 km) grid spacing. However, the current dataset (approximately 378 GB) is huge to process and store. The computational resources and disk usage required for the dataset will increase exponentially as the spatiotemporal resolution increases (Gao et al., 2018).

For such a huge amount of data, storage and extraction are not convenient. Supercomputers and parallel computing will be necessary to work with larger datasets in the future. Another limitation is that the current dataset only includes historical climate data. Many GCM products have been released, but their coarse spatial resolution and low accuracy prevent detailed projections of future climate trends and their effects on local scales, which are pressing needs for planning local strategies to cope with the negative effects of future climate changes. The Delta spatial downscaling procedure has been employed to

generate future climate data at high resolutions for some areas (Peng et al., 2017).

The issues associated with computational resources, validation, and a reasonable reference climatology must be addressed to generate high-resolution climate data for China in the future. Higher-resolution data, more validation, and a better reference climatology for historical and future climate data (version 2.0) will be concerns in future research.




**Supplement**

**Table S1:** Statistical characteristics between original/downscaled and observed monthly TMPs and PRE in the time series (1951–2016). The values shown here are the standard errors at all independent weather stations.

**Competing interests**

The authors declare that they have no conflict of interest.

**Acknowledgements**

This study was supported jointly by the National Natural Science Foundation of China (41601058 & U1703124) and the CAS Light of West China Program (XAB2015B07).

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



**Table 1:** Mean absolute errors between the observed and WorldClim climatology dataset at different spatial resolutions over the independent weather stations. The period ranges from 1970 to 2000.

|  |  | Jan | Feb | Mar | Apr | May | Jun | Jul | Aug | Sep | Oct | Nov | Dec |
|---|---|---|---|---|---|---|---|---|---|---|---|---|---|
| Minimum | 10' | 0.726 | 0.675 | 0.615 | 0.533 | 0.515 | 0.533 | 0.789 | 0.759 | 0.719 | 0.639 | 0.643 | 0.656 |
| TMP (°C) | 5' | 0.653 | 0.596 | 0.521 | 0.467 | 0.450 | 0.429 | 0.660 | 0.633 | 0.607 | 0.523 | 0.514 | 0.550 |
|  | 2.5' | 0.632 | 0.563 | 0.484 | 0.433 | 0.411 | 0.372 | 0.602 | 0.574 | 0.543 | 0.459 | 0.449 | 0.503 |
|  | 0.5' | 0.622 | 0.549 | 0.474 | 0.430 | 0.408 | 0.354 | 0.567 | 0.541 | 0.513 | 0.428 | 0.420 | 0.484 |
|  |  |  |  |  |  |  |  |  |  |  |  |  |  |
| Mean | 10' | 0.450 | 0.481 | 0.470 | 0.482 | 0.487 | 0.478 | 0.455 | 0.445 | 0.427 | 0.425 | 0.425 | 0.427 |
| TMP (°C) | 5' | 0.401 | 0.426 | 0.385 | 0.390 | 0.400 | 0.391 | 0.379 | 0.387 | 0.380 | 0.367 | 0.362 | 0.377 |
|  | 2.5' | 0.365 | 0.378 | 0.338 | 0.332 | 0.351 | 0.342 | 0.338 | 0.356 | 0.348 | 0.333 | 0.331 | 0.349 |
|  | 0.5' | 0.355 | 0.366 | 0.328 | 0.322 | 0.337 | 0.330 | 0.334 | 0.351 | 0.343 | 0.331 | 0.324 | 0.342 |
|  |  |  |  |  |  |  |  |  |  |  |  |  |  |
| Maximum | 10' | 0.832 | 0.821 | 0.809 | 0.909 | 0.827 | 0.678 | 0.718 | 0.734 | 0.644 | 0.658 | 0.630 | 0.687 |
| TMP (°C) | 5' | 0.727 | 0.711 | 0.666 | 0.760 | 0.687 | 0.560 | 0.645 | 0.658 | 0.568 | 0.561 | 0.511 | 0.576 |
|  | 2.5' | 0.664 | 0.637 | 0.591 | 0.670 | 0.597 | 0.485 | 0.589 | 0.600 | 0.531 | 0.509 | 0.447 | 0.517 |
|  | 0.5' | 0.631 | 0.596 | 0.544 | 0.611 | 0.544 | 0.445 | 0.574 | 0.578 | 0.516 | 0.484 | 0.405 | 0.479 |
|  |  |  |  |  |  |  |  |  |  |  |  |  |  |
| PRE | 10' | 2.165 | 1.869 | 3.476 | 4.662 | 5.651 | 8.416 | 9.716 | 7.993 | 5.825 | 3.968 | 2.202 | 1.378 |
| (mm) | 5' | 2.077 | 1.834 | 3.407 | 4.641 | 5.637 | 8.291 | 9.702 | 7.841 | 5.805 | 3.908 | 2.183 | 1.348 |
|  | 2.5' | 2.074 | 1.813 | 3.404 | 4.603 | 5.594 | 8.268 | 9.664 | 7.705 | 5.742 | 3.904 | 2.182 | 1.334 |
|  | 0.5' | 2.072 | 1.797 | 3.360 | 4.495 | 5.564 | 8.190 | 9.630 | 7.651 | 5.699 | 3.895 | 2.170 | 1.300 |





**Table 2:** Correlation coefficients between the observed and WorldClim climatology datasets at different spatial resolutions over the independent weather stations. The period ranges from 1970 to 2000.

| | | Jan | Feb | Mar | Apr | May | Jun | Jul | Aug | Sep | Oct | Nov | Dec |
|---|---|---|---|---|---|---|---|---|---|---|---|---|---|
| Minimum | 10' | 0.987 | 0.984 | 0.977 | 0.969 | 0.963 | 0.962 | 0.955 | 0.957 | 0.956 | 0.971 | 0.984 | 0.987 |
| TMP (°C) | 5' | 0.989 | 0.987 | 0.983 | 0.977 | 0.973 | 0.973 | 0.964 | 0.966 | 0.968 | 0.980 | 0.990 | 0.991 |
| | 2.5' | 0.989 | 0.988 | 0.985 | 0.981 | 0.978 | 0.977 | 0.968 | 0.971 | 0.974 | 0.985 | 0.992 | 0.992 |
| | 0.5' | 0.989 | 0.989 | 0.986 | 0.983 | 0.981 | 0.980 | 0.972 | 0.974 | 0.977 | 0.988 | 0.993 | 0.993 |
| | | | | | | | | | | | | | |
| Mean | 10' | 0.986 | 0.979 | 0.968 | 0.955 | 0.949 | 0.949 | 0.956 | 0.958 | 0.966 | 0.974 | 0.982 | 0.987 |
| TMP (°C) | 5' | 0.991 | 0.986 | 0.980 | 0.969 | 0.962 | 0.959 | 0.963 | 0.965 | 0.973 | 0.983 | 0.989 | 0.991 |
| | 2.5' | 0.993 | 0.990 | 0.986 | 0.977 | 0.970 | 0.965 | 0.968 | 0.970 | 0.978 | 0.986 | 0.992 | 0.993 |
| | 0.5' | 0.994 | 0.992 | 0.989 | 0.981 | 0.973 | 0.968 | 0.970 | 0.972 | 0.980 | 0.988 | 0.993 | 0.995 |
| | | | | | | | | | | | | | |
| Maximum | 10' | 0.958 | 0.946 | 0.920 | 0.892 | 0.889 | 0.899 | 0.893 | 0.890 | 0.935 | 0.957 | 0.968 | 0.974 |
| TMP (°C) | 5' | 0.969 | 0.961 | 0.946 | 0.921 | 0.912 | 0.912 | 0.898 | 0.896 | 0.939 | 0.965 | 0.978 | 0.982 |
| | 2.5' | 0.976 | 0.971 | 0.960 | 0.941 | 0.930 | 0.925 | 0.910 | 0.909 | 0.945 | 0.971 | 0.984 | 0.986 |
| | 0.5' | 0.979 | 0.976 | 0.968 | 0.951 | 0.940 | 0.932 | 0.913 | 0.912 | 0.946 | 0.973 | 0.988 | 0.989 |
| | | | | | | | | | | | | | |
| PRE | 10' | 0.976 | 0.980 | 0.978 | 0.979 | 0.974 | 0.961 | 0.903 | 0.920 | 0.941 | 0.908 | 0.939 | 0.965 |
| (mm) | 5' | 0.976 | 0.980 | 0.979 | 0.979 | 0.974 | 0.961 | 0.905 | 0.924 | 0.943 | 0.911 | 0.940 | 0.966 |
| | 2.5' | 0.976 | 0.981 | 0.980 | 0.979 | 0.974 | 0.962 | 0.908 | 0.930 | 0.943 | 0.913 | 0.941 | 0.967 |
| | 0.5' | 0.977 | 0.981 | 0.981 | 0.980 | 0.975 | 0.962 | 0.909 | 0.930 | 0.944 | 0.914 | 0.941 | 0.968 |





**Table 3:** Statistical characteristics between original/downscaled and observed monthly TMPs and PRE in the time series (1951–2016). The values shown here are the averaged evaluation results at all independent weather stations. Their standard errors are listed in Table S1.

| | Res | $MAE_c$ | $MAE_l$ | $MAE_n$ | $RMSE_c$ | $RMSE_l$ | $RMSE_n$ | $NSE_c$ | $NSE_l$ | $NSE_n$ | $Cor_c$ | $Cor_l$ | $Cor_n$ |
|---|---|---|---|---|---|---|---|---|---|---|---|---|---|
| Minimum | 30' | 1.766 | | | 1.947 | | | 0.887 | | | 0.994 | | |
| TMP (°C) | 10' | 1.673 | 1.515 | 1.558 | 1.802 | 1.726 | 1.793 | 0.896 | 0.902 | 0.899 | 0.995 | 0.995 | 0.995 |
| | 5' | 1.338 | 1.292 | 1.325 | 1.666 | 1.503 | 1.582 | 0.904 | 0.937 | 0.923 | 0.995 | 0.995 | 0.995 |
| | 2.5' | 1.233 | 1.142 | 1.211 | 1.401 | 1.349 | 1.384 | 0.946 | 0.951 | 0.949 | 0.995 | 0.997 | 0.996 |
| | 0.5' | 1.140 | 1.050 | 1.137 | 1.322 | 1.248 | 1.271 | 0.955 | 0.972 | 0.963 | 0.997 | 0.998 | 0.997 |
| Mean TMP | 30' | 1.598 | | | 1.759 | | | 0.888 | | | 0.996 | | |
| (°C) | 10' | 1.277 | 1.140 | 1.188 | 1.433 | 1.293 | 1.358 | 0.899 | 0.914 | 0.904 | 0.997 | 0.997 | 0.997 |
| | 5' | 1.117 | 0.980 | 1.003 | 1.222 | 1.133 | 1.197 | 0.926 | 0.950 | 0.933 | 0.997 | 0.997 | 0.997 |
| | 2.5' | 0.977 | 0.836 | 0.859 | 1.157 | 0.988 | 0.993 | 0.966 | 0.976 | 0.973 | 0.997 | 0.998 | 0.997 |
| | 0.5' | 0.826 | 0.820 | 0.822 | 0.974 | 0.969 | 0.970 | 0.977 | 0.981 | 0.980 | 0.998 | 0.998 | 0.998 |
| Maximum | 30' | 2.034 | | | 2.206 | | | 0.800 | | | 0.995 | | |
| TMP (°C) | 10' | 1.800 | 1.672 | 1.755 | 2.044 | 1.886 | 1.968 | 0.811 | 0.832 | 0.824 | 0.995 | 0.996 | 0.996 |
| | 5' | 1.649 | 1.487 | 1.548 | 1.864 | 1.700 | 1.756 | 0.843 | 0.856 | 0.850 | 0.996 | 0.996 | 0.996 |
| | 2.5' | 1.455 | 1.310 | 1.387 | 1.666 | 1.523 | 1.632 | 0.875 | 0.909 | 0.887 | 0.996 | 0.997 | 0.996 |
| | 0.5' | 1.296 | 1.282 | 1.291 | 1.511 | 1.491 | 1.500 | 0.909 | 0.910 | 0.910 | 0.997 | 0.997 | 0.997 |
| PRE (mm) | 30' | 17.850 | | | 29.559 | | | 0.614 | | | 0.885 | | |
| | 10' | 16.884 | 16.647 | 16.741 | 28.022 | 27.559 | 27.946 | 0.675 | 0.735 | 0.700 | 0.887 | 0.890 | 0.890 |
| | 5' | 16.134 | 15.223 | 15.942 | 26.222 | 25.185 | 25.888 | 0.764 | 0.791 | 0.773 | 0.892 | 0.900 | 0.894 |
| | 2.5' | 14.867 | 14.024 | 14.557 | 24.374 | 23.191 | 23.867 | 0.791 | 0.792 | 0.791 | 0.914 | 0.920 | 0.919 |
| | 0.5' | 13.772 | 13.269 | 13.443 | 22.655 | 21.941 | 22.213 | 0.794 | 0.808 | 0.802 | 0.920 | 0.929 | 0.926 |

Notes: Res indicates the spatial resolution. The subscripts $c$, $l$, and $n$ indicate bicubic, bilinear, and nearest-neighbor interpolations, respectively. The original TMPs and PRE are the 30' CRU data and directly compared with the observed data. Evaluations at 10', 5', 2.5', and 0.5' are the

5 evaluations for the downscaled datasets. MAE, RMSE, NSE, and Cor indicate the mean absolute error, root-mean-square error, Nash–Sutcliffe efficiency coefficient, and correlation coefficient.





**Table 4:** Comparison of the averaged climatology among the independent weather stations during 1951–2016, based on the 30' original datasets, the 0.5' downscaled datasets with the bilinear interpolation, and the observations.

|  | Monthly minimum TMP (°C) | Monthly maximum TMP (°C) | Annual mean TMP (°C) | Annual total PRE (mm) |
|---|---|---|---|---|
| 30' | -9.30 ± 0.51 | 29.55 ± 0.22 | 11.41 ± 0.30 | 898.4 ± 22.3 |
| 0.5' | -8.69 ± 0.49 | 31.27 ± 0.19 | 12.13 ± 0.28 | 879.7 ± 22.8 |
| Observation | -8.67 ± 0.51 | 31.45 ± 0.19 | 12.12 ± 0.28 | 880.2 ± 23.2 |

Notes: Monthly minimum and maximum TMPs are 1% and 99% quantile values, respectively, based on monthly time-series data. Annual total PRE and mean TMP values were calculated for full years. All values are presented as mean ± standard error.



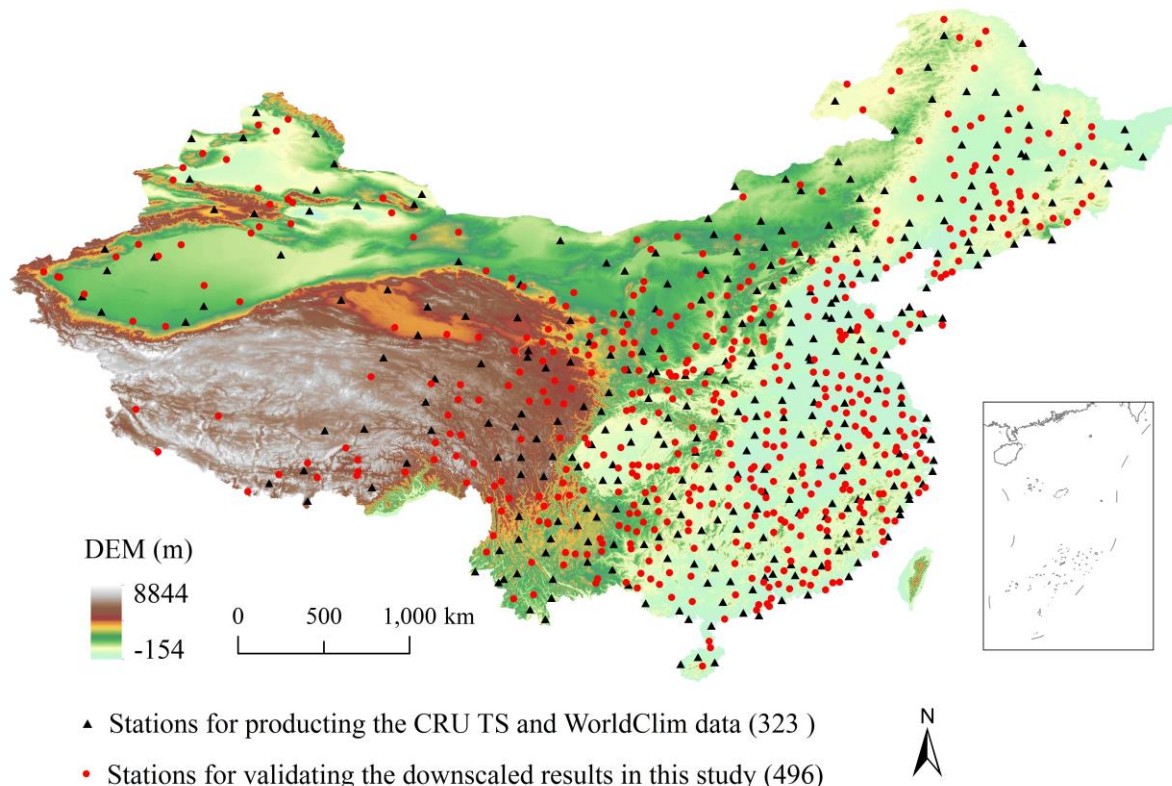

**Figure 1:** Spatial distribution of national weather stations across China.

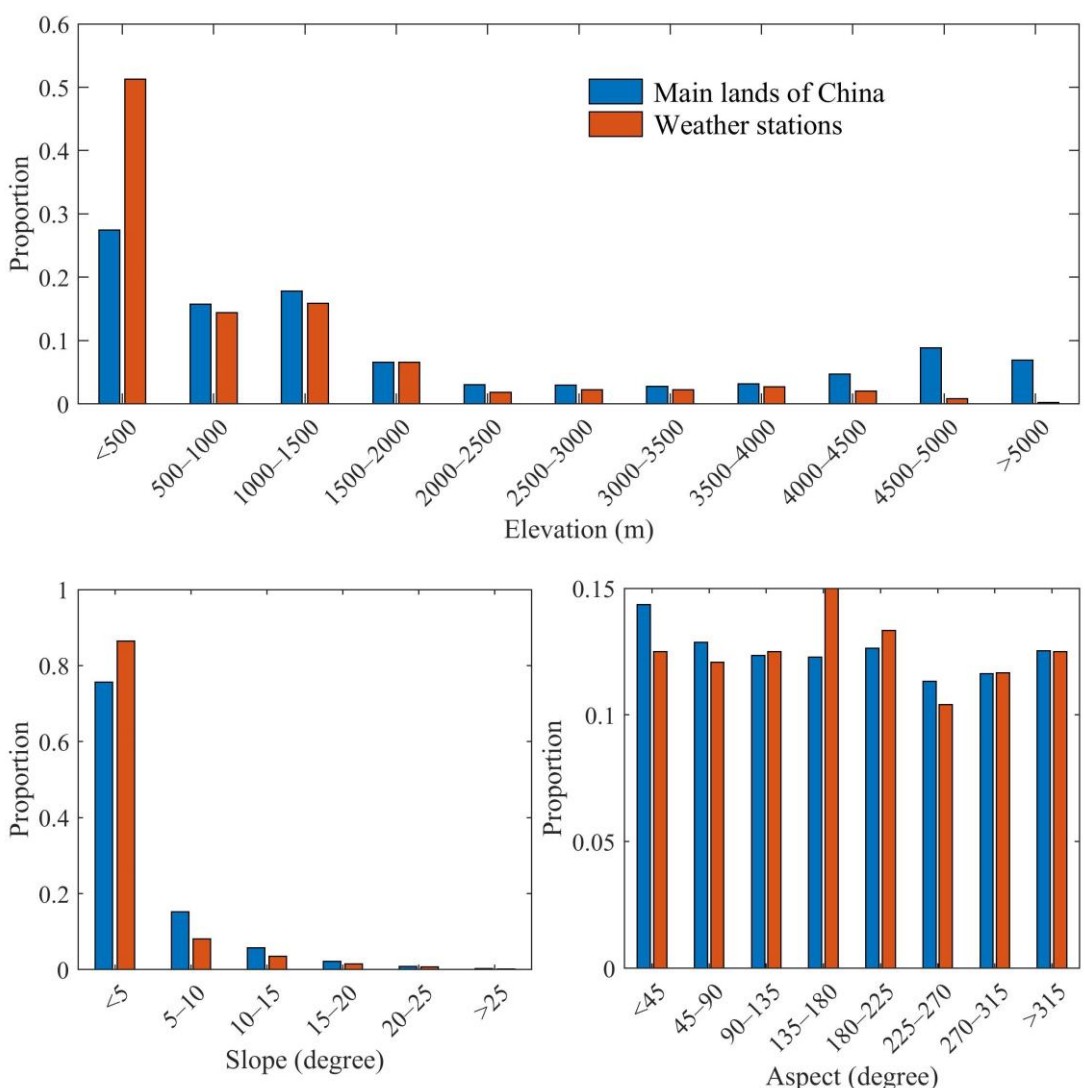

**Figure 2:** Orographic statistic information at different gradients for China and weather stations used in this study.

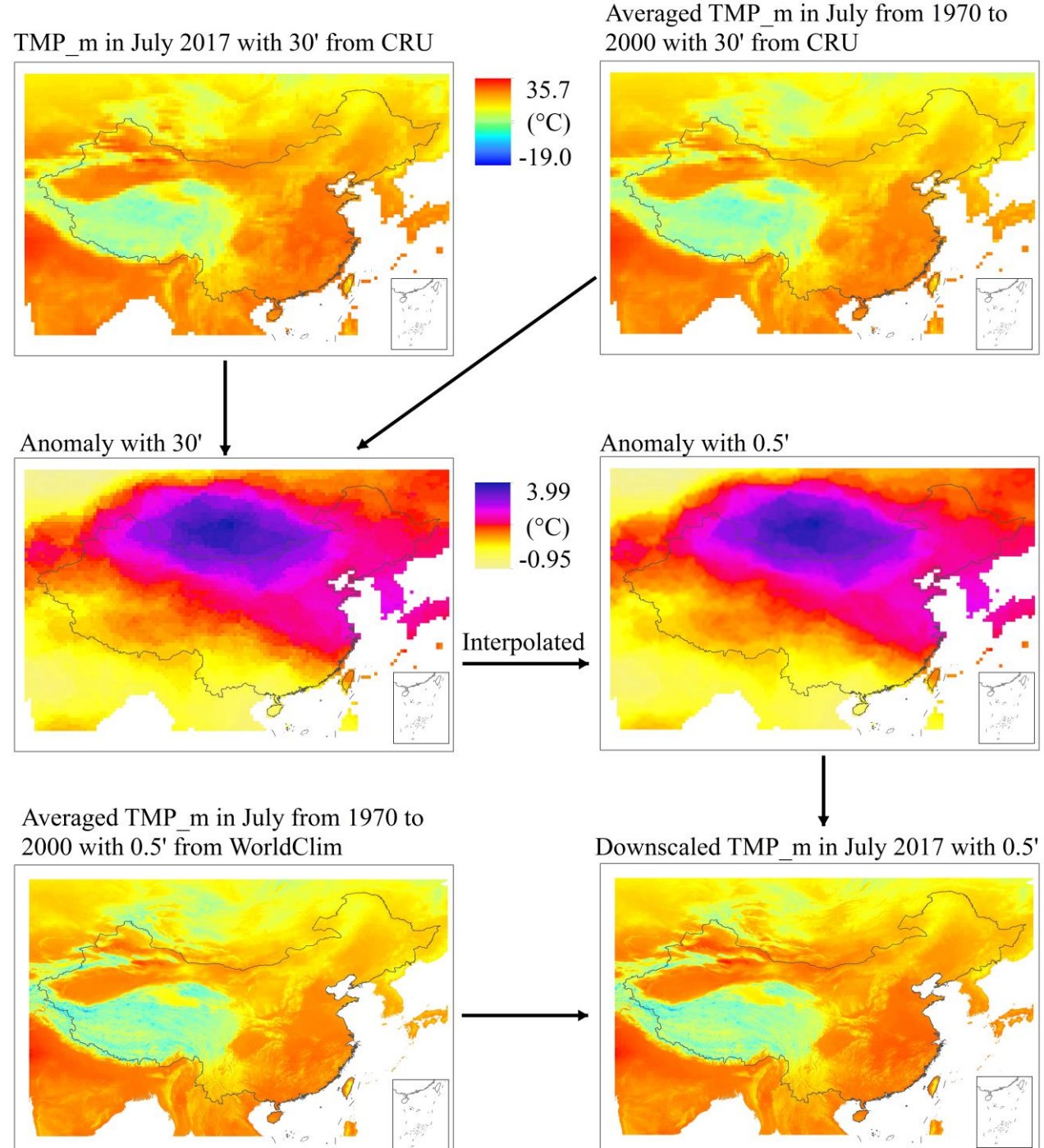

**Figure 3:** Schematic illustration of the Delta spatial downscaling process by using the mean TMP (TMP_m) in July 2017 obtained from the CRU data as an example.

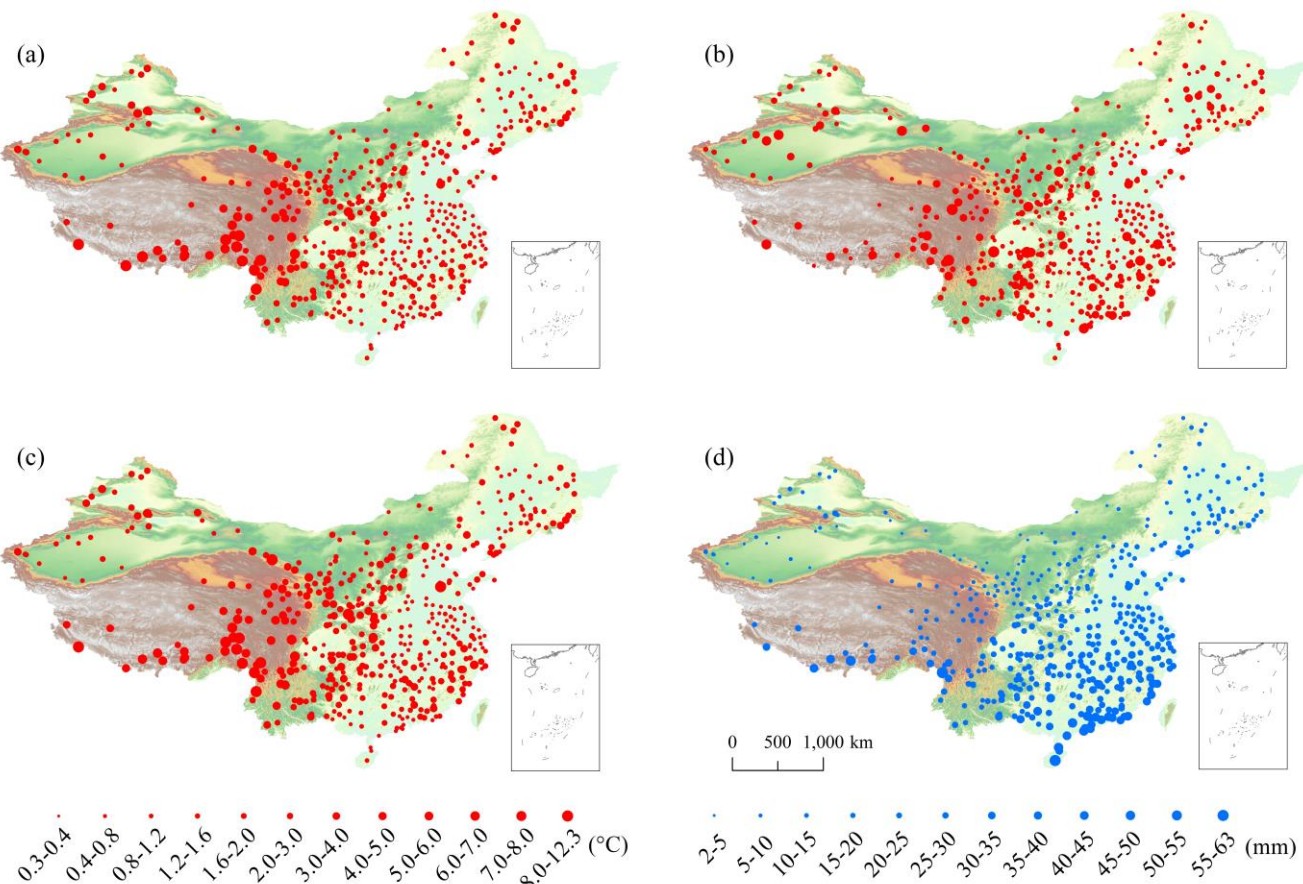

**Figure 4:** Spatial distribution of MAE between the 30' original and observed TMPs/PRE from 1951–2016 at each independent weather station. (**a**), (**b**), (**c**), and (**d**) are the MAE values for the monthly minimum, mean, and maximum temperatures as well as the monthly precipitation, respectively.



**Figure 5:** Relative decrement in MAE from the 30' original datasets to 0.5' downscaled datasets generated using bilinear interpolation at each independent weather station. (**a**), (**b**), (**c**), and (**d**) are the relative decrements in MAE in the monthly minimum, mean, and maximum temperatures as well as monthly precipitation, respectively.



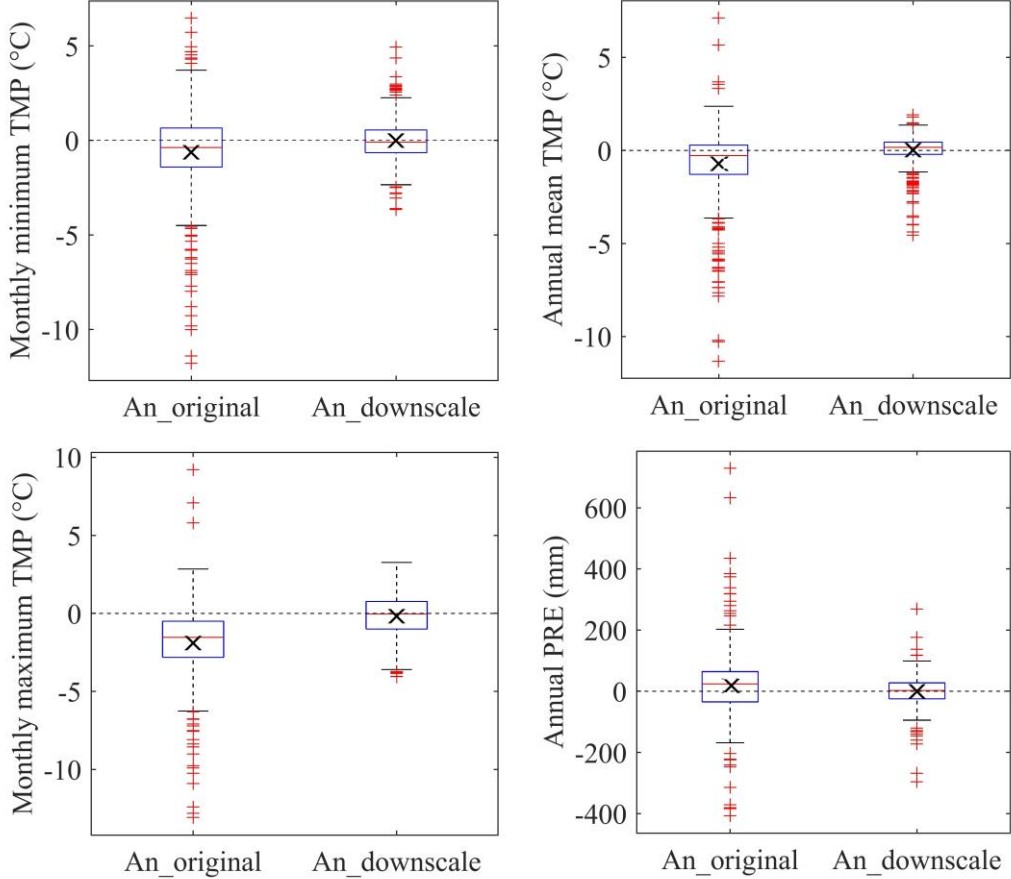

**Figure 6:** Box plots of climatology anomaly during 1951–2016 for 30' original and 0.5' downscaled datasets at independent weather stations. The climatology anomaly is equal to the bias from the original/downscaled to the observed values. The red lines in the boxes show the median values. Boxes indicate the inner-quantile range (25% to 75%). The × in the boxes indicate the averaged values of all the anomaly values. The horizontal dotted lines indicate the zero lines. The An_original and An_downscale indicate climatology anomaly of the 30' original and 0.5' downscaled datasets, respectively. The 0.5' downscaled datasets were generated using bilinear interpolation in the Delta downscaling framework. Monthly minimum and maximum TMPs are 1% and 99% quantile values, respectively, based on monthly time-series data. Annual PRE and mean TMP values were calculated for full years.



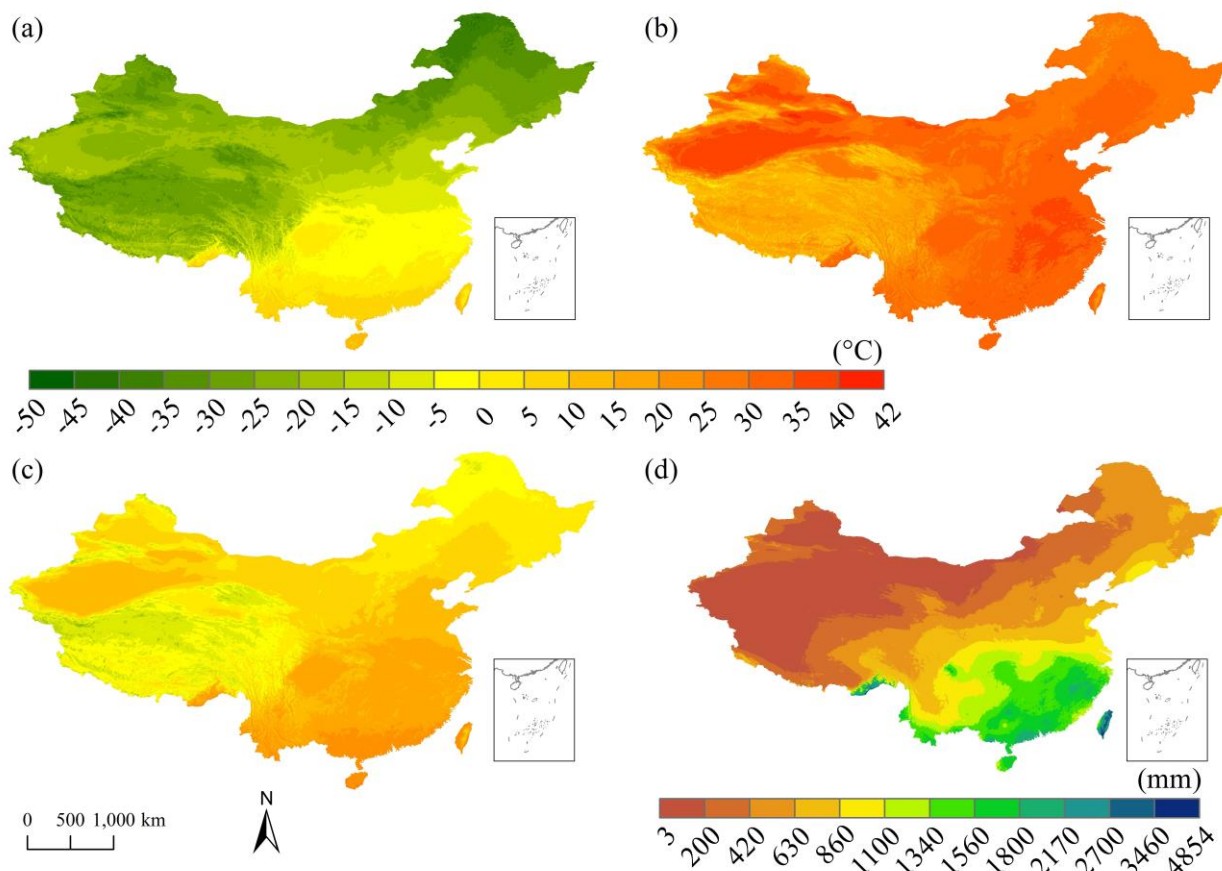

**Figure 7:** Spatial distributions of the climatology data during 1901–2017 for TMPs and PRE over China, based on the 0.5' downscaled datasets generated using bilinear interpolation in the Delta downscaling framework. (**a**) and (**b**) are the averaged annual minimum and maximum TMPs, corresponding to 1 % and 99 % quantiles of monthly minimum and maximum temperatures in a year, respectively; (**c**) and (**d**) represent the average annual mean temperature and total precipitation, respectively.

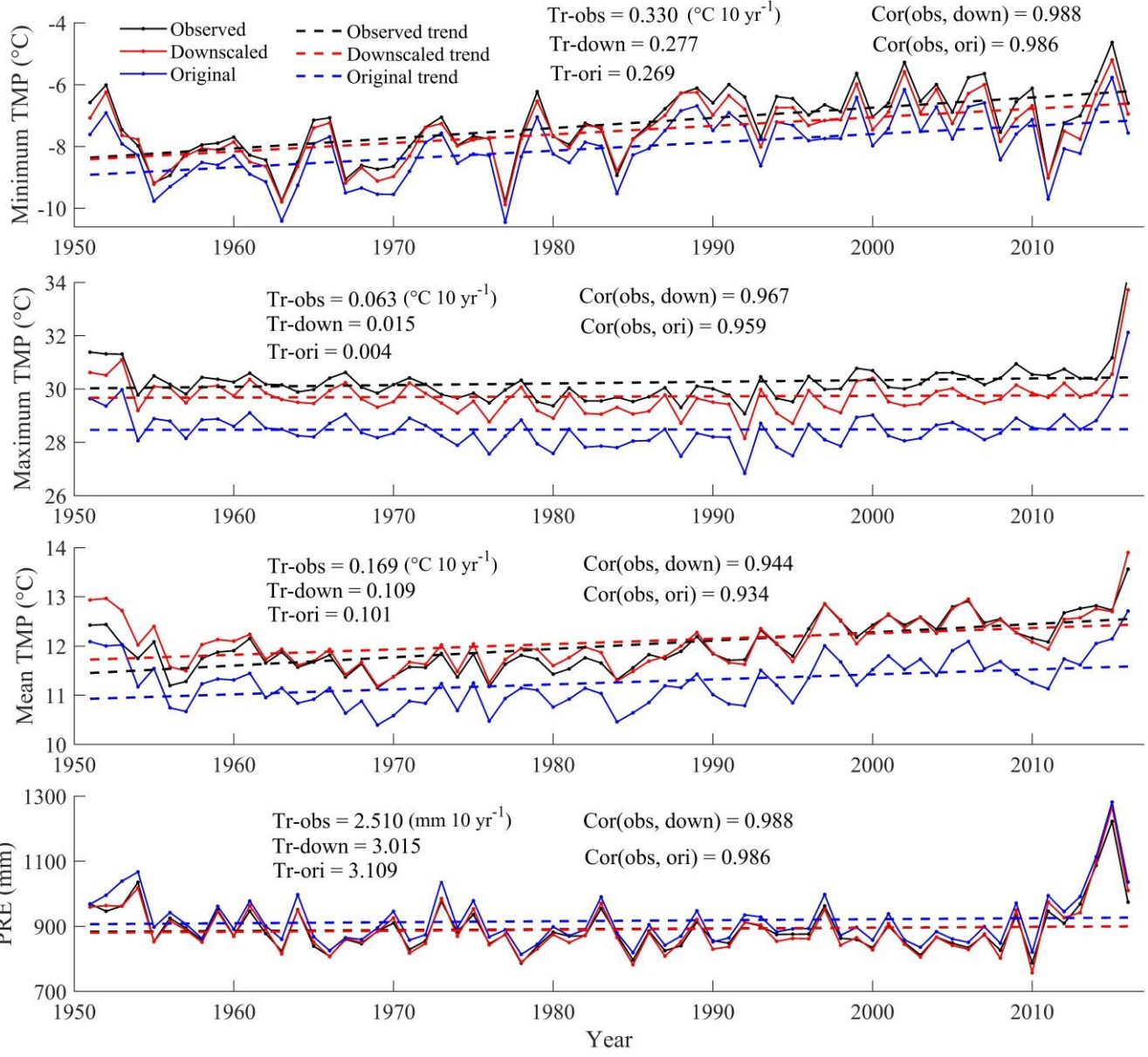

**Figure 8:** Temporal variations in annual TMPs and PRE over China during 1951–2016 based on the 0.5' downscaled datasets with bilinear interpolation, 30' original datasets, and observed datasets. The minimum and maximum TMPs are 1 % and 99 % quantiles of monthly temperatures in a year, respectively. The mean TMP and PRE are the mean of monthly temperatures and the sum of the monthly precipitations in a year, respectively. Tr-obs, Tr-down, and Tr-ori indicate the annual trends calculated using the observed, 0.5' downscaled, and 30' original datasets, respectively. Cor(obs, down) indicate the correlation coefficients of the annual values from observed and 0.5' downscaled data, while the Cor(obs, ori) indicate the correlation coefficients of the annual values from the observed and 30' original data.

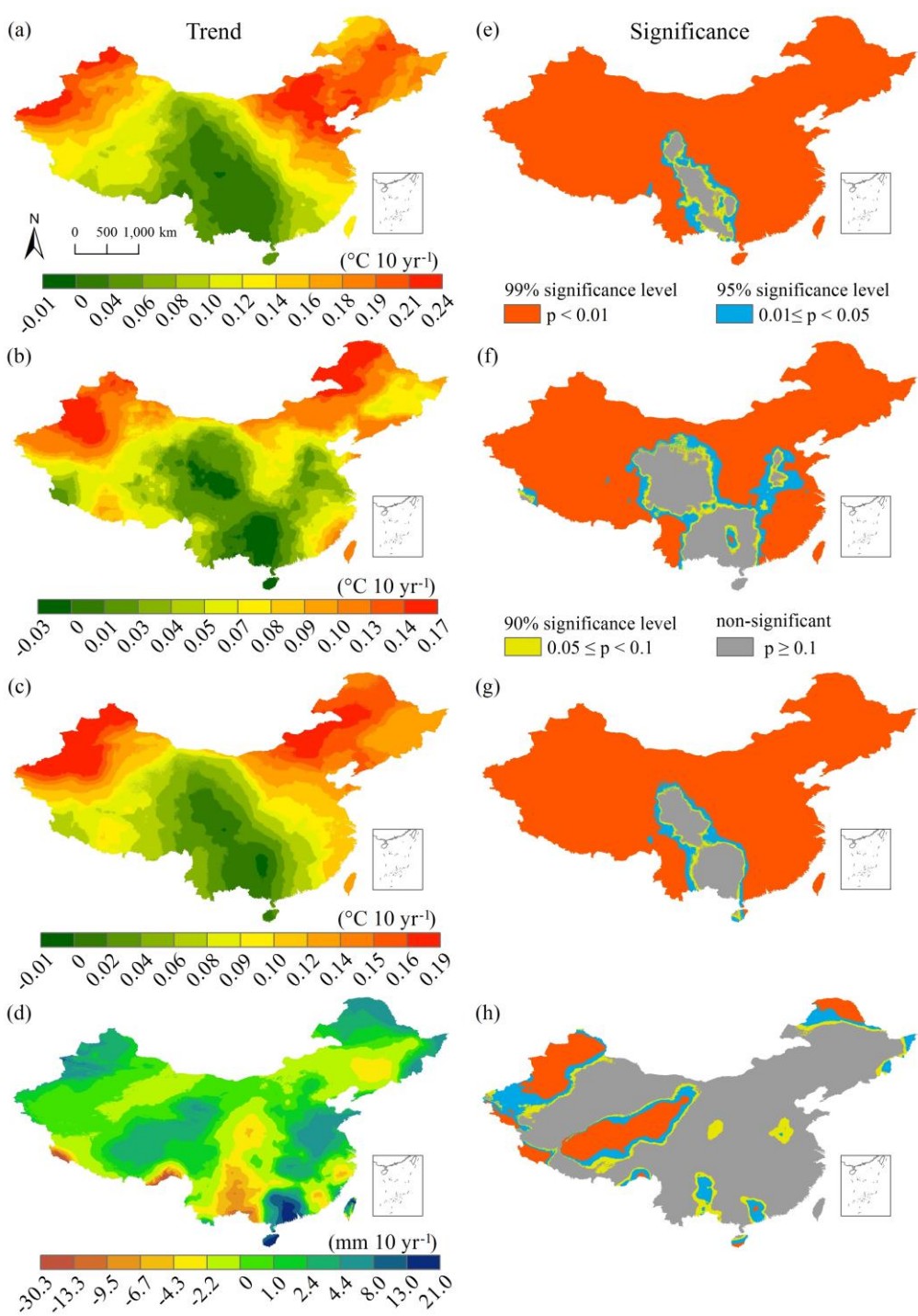

**Figure 9:** Spatial patterns of the annual trends in TMPs and PRE from 1901 to 2017 and their significance levels across China obtained using the 0.5' downscaled data with bilinear interpolation. (**ae**), (**bf**), (**cg**), and (**dh**) are the annual minimum, maximum, and mean TMPs as

well as the annual PRE, respectively. The annual minimum and maximum TMPs are 1 % and 99 % quantiles of the monthly temperatures in a year, respectively.

