# Peer review of "1-km monthly temperature and precipitation dataset for China from 1901–2017"

_Earth System Science Data, 2019_

## Referee Comment (RC1) · Anonymous Referee #1 · 9 Sep 2019

I appreciate the revision. The authors have addressed most of my concerns. However, I still find a few issues that were not clarified and I listed them below.

Major concern:

Please provide more details of how Q1 and Q99 were calculated? Is Q1 the 1% quantile of the monthly minimum TMP, and is Q99 the 99% quantile of the monthly maximum TMP? Why use the "extreme" value instead of mean monthly minimum/maximum TMP? For the results before Table 4 and Figure 6, are those minimum (maximum) TMPs the Q1 (Q99), or mean monthly minimum (maximum) TMPs?

Specific comments:

1. There are a few typos of "WordClim". The authors should check the manuscript

carefully before their resubmission.

2. P1,L23: "the downscaling procedure used data from CRU and WordClim and did not incorporate observations" - I may understand what the authors mean, but this statement is not accurate. First, CRU and WorldClim have already incorporated observations. Second, if their "observations" mean the observations from 496 weather stations, the downscaling procedure itself never uses any of those station observations even for the period 1951-2016. The quality of the new datasets throughout the period 1901-2017 depends on the quality of the CRU and WorldClim datasets.

3. P1, L25: The authors should mention the evaluation of the input data (CRU and WorldClim) at the beginning, then discuss the quality of their downscaling products.

4. P6, L10: Evaluation criteria -> Evaluation metrics

5. P6, L20: is n the number of stations, or the number of months? I assume it is about the number of stations, so Pi is the climatology of the original or downscaled values, and Oi is the climatology of observed values? Because the authors mentioned "time series" in the results, this information should be clarified to avoid confusion.

6. P7, L16: What does "averaged evaluation" mean? Do the authors mean "evaluation of climatology of the downscaled monthly TMPs and PRE averaged over independent weather stations"?

7. P9,L13: time correlation -> temporal correlation

8. Figure 9: add hatching on the trend map to show the significance instead of plotting individual significance maps

---

## Referee Comment (RC2) · Anonymous Referee #2 · 12 Sep 2019

The manuscript by Peng et al presents a valuable monthly climatic dataset across China by downscaling CRU data. This dataset has a high spatial resolution and cover a long time period. The evaluations by comparing this dataset with WorldClim data at different spatial resolution et al have demonstrated that the new dataset is reasonable. The analysis of climatology and annual trend further show that the new dataset could be used for investigations related to climate change across China.

Overall, the paper is very well written and the data would be very useful for scientific communities. Here I have some specific suggestions to improve this paper.

Specific Comments 1. In the introduction, it may be useful to mention or introduce ERA5 climate data, which has very high time resolution (hourly data) 2. P6, 3.2 Evaluation criteria. This part focused on the evaluation of original/downscaled dataset.

Authors should present a brief description for the evaluation of WorldClim data. 3. P8, L9. Authors introduced the climatic variables for the climatology analysis in the Result section. These introductions should be placed in the Method section. 4. P27, Figure 8. Authors used the 1% and 99% quantiles of monthly temperatures in a year to represent the annual minimum and maximum temperatures. Why didn't use the minimum (maximum) value of monthly minimum (maximum) temperatures in a year to indicate the annual minimum (maximum) temperature? Although they should be the same values as my experience, I think that the later is more widely used. 5. P28, Figure 9. The significance levels in the right column should be integrated into the trends the left column, using recognizable lines. Besides, in the P9 L16, authors have stated the 95% significance level was adopted for the significance. Thus, only the 95% Significance level in the Figure 9 should be presented. 6. Authors should carefully check the typos, such as "WordClim" in P1 L24 25,

Please also note the supplement to this comment:
https://www.earth-syst-sci-data-discuss.net/essd-2019-145/essd-2019-145-RC2-supplement.pdf

---

## Author Comment (AC1) · 29 Oct 2019

I appreciate the revision. The authors have addressed most of my concerns. However, I still find a few issues that were not clarified and I listed them below.

Major concern:

Please provide more details of how Q1 and Q99 were calculated? Is Q1 the 1% quantile of the monthly minimum TMP, and is Q99 the 99% quantile of the monthly maximum TMP? Why use the "extreme" value instead of mean monthly minimum/maximum TMP? For the results before Table 4 and Figure 6, are those minimum (maximum) TMPs the Q1 (Q99), or mean monthly minimum (maximum) TMPs?

Response: Many thanks for your queries. Indeed, Q1 is the 1% quantile of the monthly minimum TMP in time series, and Q99 is the 99% quantile of the monthly maximum TMP in time series. In the manuscript, we want to use them to present the extreme minimum and maximum TMP. However, after we see your queries, we think the "Q1" and "Q99" are not suitable for the monthly TMPs, because they are often used in the analysis of the daily TMP.

For a reasonable representation of annual climatology in minimum/maximum TMP in this study, we will use the mean annual minimum and maximum TMPs. Specifically, annual minimum TMP is the minimum value of the monthly minimum TMPs in a year, and annual maximum TMP is the maximum value of the monthly maximum TMPs in a year.

Compared with the mean monthly minimum/maximum TMP in a year, the minimum value of monthly minimum TMPs and the maximum value of monthly maximum TMPs in a year could more represent their annual climatology.

We have recalculated the results of climatology in minimum/maximum TMP and revised the Table 4 and Figures 6-7. The implications of these results are the same as the meaning mentioned in the previous manuscript. According to the new results, we have revised the related contents as following.

Page 8 Lines 27-29 "*Specifically, the averaged climatology differences between the 0.5' downscaled and observed data equaled −0.12 °C for the annual minimum TMP, −0.12 °C for the annual maximum TMP, 0.01 °C for the annual mean TMP, and −0.5 mm for the annual total PRE.*"

Page 9 Lines 6-9 "*The mean annual minimum TMP for China ranged from −47.44 to 18.70 °C, with an average of −13.19 °C, and the lowest value corresponded to a location in the western part of the Qinghai–Tibet Plateau (Fig. 7a). The mean annual maximum TMP ranged from −17.53 to 42.23 °C, with an average of 26.70 °C, and the highest value was observed at a location in the Turpan Basin (Fig. 7b).*"

In addition, the annual trend analysis of minimum/maximum TMP employed the "Q1" and "Q99" in a year for presenting their annual values. We have recalculated the their annual values as above. The results show that the minimum value of monthly minimum TMPs in a year is equal to its "Q1", and the maximum value of monthly maximum TMPs in a year is equal to its "Q99". Thus, annual trend analysis results of minimum/maximum TMP are reasonable, and we have clarified the related introduction of how to calculate the annual minimum/maximum TMP in Page 7 Lines 4-7.

*"Specifically, the annual minimum TMP was the minimum value of monthly minimum TMPs in a year, the annual maximum TMP was the maximum value of the monthly maximum TMPs in a year, the annual mean TMP was the mean of the monthly mean TMPs in a year, and the annual PRE was the sum of the monthly precipitations in a year."*

Specific comments:

1. There are a few typos of "WordClim". The authors should check the manuscript carefully before their resubmission.

Response: Many thanks for your attentions. We have revised the typos in the revision.

2. P1, L23: "the downscaling procedure used data from CRU and WordClim and did not incorporate observations" - I may understand what the authors mean, but this statement is not accurate. First, CRU and WorldClim have already incorporated observations. Second, if their "observations" mean the observations from 496 weather stations, the downscaling procedure itself never uses any of those station observations even for the period 1951-2016. The quality of the new datasets throughout the period 1901-2017 depends on the quality of the CRU and WorldClim datasets.

Response: Yes, you are right. We have revised this statement in Page 1 Lines 27-31.

*"Although the new dataset was not evaluated before 1950 owing to data unavailability, the quality of the new dataset in the period of 1901–2017 depended on the quality of the original CRU and WorldClim datasets. Therefore, the new dataset was reliable, as the downscaling procedure further improved the quality and spatial resolution of the CRU dataset, and was concluded to be useful for investigations related to climate change across China."*

3. P1, L25: The authors should mention the evaluation of the input data (CRU and WorldClim) at the beginning, then discuss the quality of their downscaling products.

Response: Thanks for your suggestion. We have revised the Abstract in Page 1.

*"High-spatial-resolution and long-term climate data are highly desirable for understanding climate-related natural processes. China covers a large area with a low density of weather stations in some (e.g., mountainous) regions. This study describes a 0.5' (~1-km) dataset of monthly air temperatures at 2 m (minimum, maximum, and mean TMPs) and precipitation (PRE) for China in the period of 1901–2017. The dataset was spatially downscaled from the 30' climatic research unit (CRU) time series dataset with the climatology dataset of WorldClim using Delta spatial downscaling and evaluated using observations collected in 1951–2016 by 496 weather stations across China. Prior to downscaling, we evaluated the performances of the WorldClim data with different spatial resolutions and the 30' original CRU dataset using the observations, revealing that their qualities were overall satisfactory. Specifically, WorldClim data exhibited better performance at higher spatial resolution, while the 30' original CRU dataset had low biases and high performances. Bicubic, bilinear, and nearest-neighbor interpolation methods employed in downscaling processes were compared, and bilinear interpolation was found to exhibit the best performance to generate the downscaled dataset. Compared with the evaluations of the 30' original CRU dataset, the mean absolute error of the new dataset (i.e., of the 0.5' dataset downscaled by bilinear*

*interpolation) decreased by 35.4–48.7 % for TMPs and by 25.7 % for PRE, the root-mean-square error decreased by 32.4–44.9 % for TMPs and by 25.8 % for PRE, the Nash–Sutcliffe efficiency coefficients increased by 9.6–13.8 % for TMPs and by 31.6 % for PRE, and correlation coefficients increased by 0.2–0.4 % for TMPs and by 5.0 % for PRE. The new dataset could provide detailed climatology data and annual trends of all climatic variables across China, and the results could be well evaluated using observations at the station. Although the new dataset was not evaluated before 1950 owing to data unavailability, the quality of the new dataset in the period of 1901–2017 depended on the quality of the original CRU and WorldClim datasets. Therefore, the new dataset was reliable, as the downscaling procedure further improved the quality and spatial resolution of the CRU dataset, and was concluded to be useful for investigations related to climate change across China. The dataset presented in this article has been published in the Network Common Data Form (NetCDF) at http://doi.org/10.5281/zenodo.3114194 for precipitation (Peng, 2019a) and http://doi.org/10.5281/zenodo.3185722 for air temperatures at 2 m (Peng, 2019b) and includes 156 NetCDF files compressed in zip format and one user guidance text file."*

4.   P6, L10: Evaluation criteria -> Evaluation metrics

Response: Adopted.

5.   P6, L20: is n the number of stations, or the number of months? I assume it is about the number of stations, so Pi is the climatology of the original or downscaled values, and Oi is the climatology of observed values? Because the authors mentioned "time series" in the results, this information should be clarified to avoid confusion.

Response: Many thanks for your attentions. This study evaluated the original CRU and downscaled datasets in the time series, and evaluated the WorldClim data in the geographic space. The evaluation presented here is for the original CRU and downscaled datasets. We have revised this section and added the introduction of how to evaluate the WorldClim data in Page 6 Lines 22-28.

*"where $P_i$ is the original or downscaled value in the time series, $O_i$ is the observed value in the time series, and n is the number of months. Evaluations of the original CRU and downscaled datasets were carried out at each independent station to be mapped in geographic space, and the obtained results were averaged over all independent stations to compare the overall performances of original CRU and downscaled datasets.*

*   In addition, WorldClim data at different spatial resolutions were evaluated using MAE and Cor indices, which were calculated according to the paired climatology values from WorldClim and observed data for the same geographic position. The sample size was the number of independent stations."*

6.   P7, L16: What does "averaged evaluation" mean? Do the authors mean "evaluation of climatology of the downscaled monthly TMPs and PRE averaged over independent weather stations"?

Response: Many thanks for your queries. This section presented the evaluations of downscaled datasets. The evaluations of the downscaled datasets were carried out at each independent station, and then averaged over all independent stations. We have revised this sentence in Page 8 Lines 2-3.

"*Table 3 presents the averaged evaluation over independent weather stations, based on the evaluation result at each station for the downscaled monthly TMPs and PRE in the time series (1951–2016) at different spatial resolutions.*"

7. P9, L13: time correlation -> temporal correlation

Response: Adopted.

8. Figure 9: add hatching on the trend map to show the significance instead of plotting individual significance maps.

Response: Adopted. We have revised the Figure 9 as following.

[Figure]

**Figure 9:** Spatial patterns of the annual trends in TMPs and PRE from 1901 to 2017 across China obtained using the 0.5' downscaled data with bilinear interpolation. (**a**)–(**d**) correspond to the annual minimum, maximum, and mean TMPs as well as the annual PRE, respectively. Purple zones indicate locations where trends are significant at the 95 % confidence level.

After above revisions, we found a professional English editor to improve the language quality of manuscript.

[revised manuscript text omitted]

---

## Author Comment (AC2) · 29 Oct 2019

The manuscript by Peng et al presents a valuable monthly climatic dataset across China by downscaling CRU data. This dataset has a high spatial resolution and cover a long time period. The evaluations by comparing this dataset with WorldClim data at different spatial resolution et al have demonstrated that the new dataset is reasonable. The analysis of climatology and annual trend further show that the new dataset could be used for investigations related to climate change across China.

Overall, the paper is very well written and the data would be very useful for scientific communities. Here I have some specific suggestions to improve this paper.

Specific Comments

1. In the introduction, it may be useful to mention or introduce ERA5 climate data, which has very high time resolution (hourly data).

Response: Thanks for your suggestion. Although the ERA5 climate data has a very high time resolution, available ERA5 climate data starts from 1979. This study focused on the generation of long term monthly climate data, and thus several > 100 years monthly climate datasets were reviewed in the Introduction section. Accordingly, we insist the original statement for a clear understanding. Based on your suggestion, we will focus on the downscaling of ERA5 climate data to a high spatial resolution in our future work for a generation of high-time and spatial resolution dataset. Again, we appreciate your suggestion very much.

2. P6, 3.2 Evaluation criteria. This part focused on the evaluation of original/downscaled dataset. Authors should present a brief description for the evaluation of WorldClim data.

Response: Thanks for your suggestion. We have added a brief description in Page 6 Lines 26-28.

"*In addition, WorldClim data at different spatial resolutions were evaluated using MAE and Cor indices, which were calculated according to the paired climatology values from WorldClim and observed data for the same geographic position. The sample size was the number of independent stations.*"

3. P8, L9. Authors introduced the climatic variables for the climatology analysis in the Result section. These introductions should be placed in the Method section.

Response: Thanks for your suggestion. We have added a subsection in the Method to describe how to evaluate the climatology and trends for the downscaled dataset and introduce the related variables in Page 7 Line 1.

"*3.3 Evaluations of climatology and trends for the downscaled dataset*

*We also evaluated the climatology and trends for the 0.5' downscaled dataset by comparison with the 30' original CRU and observed datasets. The mean annual value of each climatic variable was used to represent climatology, and the annual trend was employed to indicate temporal variation. Specifically, the annual minimum TMP was the minimum value of monthly minimum TMPs in a year, the annual maximum TMP was the maximum value of the monthly maximum TMPs in a year, the annual mean TMP was the mean of the monthly mean TMPs in a year, and the annual PRE was the sum of the monthly*

*precipitations in a year. For annual trend analysis, linear regression relationships between climatic variables and year were established to calculate the trend magnitude."*

4. P27, Figure 8. Authors used the 1% and 99% quantiles of monthly temperatures in a year to represent the annual minimum and maximum temperatures. Why didn't use the minimum (maximum) value of monthly minimum (maximum) temperatures in a year to indicate the annual minimum (maximum) temperature? Although they should be the same values as my experience, I think that the later is more widely used.

Response: Yes, you are right. We recalculated the annual minimum/maximum temperature as your suggestion. The results show that they are the same values in a year. We have revised the related statement for these two variables throughout the text as your suggestion.

5. P28, Figure 9. The significance levels in the right column should be integrated into the trends the left column, using recognizable lines. Besides, in the P9 L16, authors have stated the 95% significance level was adopted for the significance. Thus, only the 95% Significance level in the Figure 9 should be presented.

Response: We have revised Figure 9 as your suggestion. The revised Figure is as following.

[Figure]

**Figure 9:** Spatial patterns of the annual trends in TMPs and PRE from 1901 to 2017 across China obtained using the 0.5' downscaled data with bilinear interpolation. (**a**)–(**d**) correspond to the annual minimum, maximum, and mean TMPs as well as the annual PRE, respectively. Purple zones indicate locations where trends are significant at the 95 % confidence level.

6. Authors should carefully check the typos, such as "WordClim" in P1 L24 25,

Response: Many thanks for your attention. We have corrected the typos throughout the text.

After above revisions, we found a professional English editor to improve the language quality of manuscript.

[revised manuscript text omitted]